# *DLM-3D*: Diffusion Language Models for 3D Point Clouds Generation

## Abstract

Generating high-fidelity and diverse 3D point clouds is a fundamental challenge in 3D vision. Prior approaches primarily rely on autoregressive models or continuous diffusion processes, which often suffer from limited scalability, slow inference, and difficulties in modeling long-range dependencies across unordered point sets. In this work, we introduce *DLM-3D*, the first framework that adapts diffusion language models to the domain of 3D shape generation. Our key idea is to tokenize 3D point clouds into discrete semantic units and leverage discrete diffusion denoising over this sequence space, enabling parallel generation while preserving geometric fidelity. To better capture the intrinsic structure of point clouds, we design a permutation-invariant tokenizer and a geometry-aware noise schedule, which together allow *DLM-3D* to learn both local geometric consistency and global shape coherence. Extensive experiments on ShapeNet and ModelNet demonstrate that *DLM-3D* achieves state-of-the-art performance in terms of fidelity, diversity, and coverage, significantly outperforming autoregressive and continuous diffusion baselines. Moreover, *DLM-3D* supports flexible generation modes, including shape completion and conditional synthesis, without task-specific retraining.

## 1 Introduction

3D point cloud generation has become a central task in computer vision and graphics, with applications spanning 3D content creation, virtual reality, autonomous driving, and embodied AI. The ability to synthesize high-quality and diverse point clouds is essential not only for downstream perception and reconstruction tasks, but also for enabling interactive and scalable 3D environments. However, generating realistic 3D shapes remains a challenging problem due to the unordered nature of point sets and the need to capture both local geometric details and global structural coherence.

Existing generative approaches (Mo et al., 2023a;b; Liu et al., 2023b) can be broadly categorized into autoregressive models and continuous diffusion models. Autoregressive models discretize 3D structures into sequences and model them token-by-token, but they suffer from slow inference, exposure bias, and error accumulation, while struggling to capture long-range dependencies across large point sets. On the other hand, continuous diffusion models treat 3D point clouds as continuous signals and learn to denoise corrupted coordinates. Despite their success in generating realistic shapes, such models are computationally expensive, sensitive to sampling hyperparameters, and often fail to scale to large, diverse datasets. Moreover, they typically require hundreds of denoising steps, which limits practical deployment.

A core difficulty in 3D point cloud generation lies in bridging the gap between discrete representation learning and the unordered geometry of 3D shapes. Unlike text or images, point clouds lack a natural spatial ordering, which makes sequence modeling ambiguous and prone to bias. At the same time, the model must capture long-range global structure (*e.g.*, overall symmetry, topology) while faithfully reconstructing fine-grained local geometry (*e.g.*, surface details, thin parts). Balancing these two levels of structure is inherently difficult. Efficiency poses another challenge: autoregressive methods are bottlenecked by sequential decoding, while continuous diffusion requires dense sampling in Euclidean space. Finally, flexibility remains limited as most prior models are tailored to unconditional generation and cannot seamlessly adapt to tasks like completion or conditional synthesis.

In this work, we introduce *DLM-3D*, the first framework that adapts *diffusion language models* to the domain of 3D point cloud generation. Our key insight is that discrete diffusion, originally designed for

language modeling, can naturally handle unordered data when combined with permutation-invariant tokenization. Specifically, we tokenize point clouds into discrete semantic units that encode both geometry and neighborhood context, ensuring order-agnostic representations. We then apply discrete diffusion denoising in the token space, which enables parallel refinement of multiple tokens per step and avoids the inefficiency of autoregressive decoding. To further exploit geometric structure, we design a *geometry-aware noise schedule* that perturbs tokens in proportion to their structural importance, preserving local neighborhoods and stabilizing global coherence during denoising. This combination allows DLM-3D to scale to large datasets, generate in parallel, and support flexible conditioning (*e.g.*, completion from partial scans) without retraining. In essence, *DLM-3D* unifies the efficiency of discrete diffusion with the structural inductive biases of 3D geometry.

We conduct extensive experiments on ShapeNet and ModelNet40 to evaluate the effectiveness of *DLM-3D*. Our approach consistently achieves state-of-the-art performance across fidelity, diversity, and coverage metrics, surpassing both autoregressive and continuous diffusion baselines. Beyond class-conditional generation, *DLM-3D* demonstrates remarkable flexibility: the same trained model supports point cloud completion, conditional generation, interpolation, and inpainting, all without task-specific retraining. These results highlight the scalability, robustness, and versatility of diffusion language modeling for 3D point cloud generation.

Overall, we summarize our contributions below:

- We introduce *DLM-3D*, the first framework that adapts diffusion language models to unordered point clouds, enabling efficient and high-fidelity generation in discrete token space.

- We propose a *permutation-invariant tokenizer* and a *geometry-aware noise schedule*, which together preserve local geometric details, maintain global coherence, and ensure robustness to input order and incomplete observations.

- We demonstrate state-of-the-art results on ShapeNet and ModelNet40 benchmarks, and show that *DLM-3D* naturally supports a wide range of applications including completion, conditional generation, interpolation, and inpainting, without task-specific retraining.

- We provide extensive ablations and theoretical analysis validating the effectiveness of our design choices, highlighting efficiency, scalability, and robustness advantages over autoregressive and continuous diffusion baselines.

## 2 RELATED WORK

**Diffusion Language Models.** Diffusion language models (DLMs) extend the diffusion paradigm from continuous domains to discrete sequences. Instead of modeling data in coordinate space, DLMs operate over discrete tokens, progressively corrupting and denoising categorical representations. Recent works have applied this framework to text and code, demonstrating advantages over autoregressive models in terms of efficiency, parallel decoding, and robustness to exposure bias (Austin et al., 2021; Hoogeboom et al., 2021; Liu et al., 2023a). By combining the strengths of diffusion and language modeling, DLMs enable scalable sequence generation with fewer denoising steps than continuous diffusion and without the strict left-to-right constraints of autoregression. Our work builds on this paradigm but extends it into the 3D domain, where unordered point sets require specialized tokenization and geometry-aware diffusion to capture both local and global structure.

**Diffusion Models for Point Clouds Generation.** Diffusion models (Ho et al., 2020; Song et al., 2021b;a) have achieved state-of-the-art performance in images (Saharia et al., 2022), videos (Ho et al., 2022), and speech (Kong et al., 2021), leveraging iterative denoising to synthesize high-quality samples. Extending diffusion to 3D, several methods operate directly in coordinate space. PVD (Zhou et al., 2021) introduced probabilistic diffusion for point clouds, while LION (Zeng et al., 2022) incorporated latent priors to improve shape fidelity. MeshDiffusion (Liu et al., 2023b) and related works extend diffusion to mesh representations. Although effective, these approaches suffer from high computational cost, sensitivity to sampling schedules, and difficulty scaling to diverse object categories due to their reliance on continuous-space denoising. Our approach differs fundamentally by performing diffusion in the discrete token space, which reduces sampling cost and enables parallel refinement while preserving geometric fidelity.

**Autoregressive and Transformer-Based Generative Models.** Beyond diffusion, autoregressive and transformer-based models have been explored for 3D point cloud generation. Early works such as r-GAN and l-GAN (Achlioptas et al., 2018) applied GANs to point sets, while PointFlow (Yang et al., 2019) and SetVAE (Kim et al., 2021) explored normalizing flows and variational approaches. More recently, DiT-3D (Mo et al., 2023a) and FastDiT-3D (Mo et al., 2023b) scaled transformer-based diffusion to large datasets, demonstrating strong performance. However, autoregressive approaches suffer from slow decoding and error accumulation, while transformer-based continuous diffusion requires hundreds of denoising steps. In contrast, `DLM-3D` leverages permutation-invariant tokenization and discrete diffusion, achieving superior fidelity and diversity with significantly fewer diffusion steps.

## 3 METHOD

In this section, we present `DLM-3D`, our framework for 3D point cloud generation using diffusion language models. We begin with preliminaries and notations, then introduce the two core components: (i) a permutation-invariant tokenizer that discretizes 3D point clouds into semantic tokens, and (ii) a discrete diffusion process with a geometry-aware noise schedule. Finally, we describe the denoising transformer backbone and training objectives.

### 3.1 PRELIMINARIES

**Problem Setup and Notations.** We consider a point cloud $\mathbf{X} = \{\mathbf{x}_i\}_{i=1}^N$ where each $\mathbf{x}_i \in \mathbb{R}^3$ denotes the 3D coordinates of a point. The generative goal is to learn a model $p_\theta(\mathbf{X})$ that approximates the underlying distribution of real-world shapes, while supporting conditional variants such as completion and class-conditional synthesis. To adapt diffusion language modeling, we introduce a tokenizer $f_\phi : \mathbb{R}^{N \times 3} \to \{1, \ldots, K\}^M$ that maps the unordered point set into a discrete token sequence $\mathbf{z} = [z_1, \ldots, z_M]$ of length $M$, drawn from a codebook of size $K$. The token sequence $\mathbf{z}$ serves as the input space for our discrete diffusion process.

### 3.2 PERMUTATION-INVARIANT TOKENIZATION

A central challenge in modeling 3D point clouds is their lack of inherent ordering. Standard sequence tokenizers introduce artificial biases by enforcing arbitrary point orderings, which can lead to poor generalization and unstable training. To address this, we design a *permutation-invariant tokenizer* that learns discrete codes from local neighborhoods while remaining agnostic to input permutations.

**Local patch extraction.** Given a point cloud $\mathbf{X} = \{\mathbf{x}_i\}_{i=1}^N$, we select $M$ anchor points using farthest-point sampling (FPS) to ensure uniform coverage across the shape. For each anchor, we gather a local patch using $k$-nearest neighbors in Euclidean space. This balances dense and sparse regions, capturing both global structure and fine details.

**Patch encoding and quantization.** Each patch $\mathcal{P}_j \subset \mathbb{R}^{k \times 3}$ is encoded by a shared PointNet-style encoder $h_\phi$, yielding latent embeddings $\mathbf{e}_j = h_\phi(\mathcal{P}_j) \in \mathbb{R}^d$. These embeddings are then quantized via vector quantization (VQ) against a learned codebook $\mathcal{C} = \{\mathbf{c}_1, \ldots, \mathbf{c}_K\}$:

$$z_j = \arg \min_{k \in \{1, \ldots, K\}} \|\mathbf{e}_j - \mathbf{c}_k\|_2.$$

The sequence $\mathbf{z} = [z_1, \ldots, z_M]$ forms the discrete representation of the shape.

**Permutation invariance.** By associating tokens with anchor patches rather than individual points, the resulting sequence is independent of point ordering. The anchor index provides a stable spatial reference, while learned embeddings capture local geometry. Compared to naive rasterization or voxelization, this approach yields compact sequences ($M \ll N$) while preserving structural information.

**Benefits.** This design offers: (i) robustness to permutations and resampling, (ii) compression of raw coordinates into semantically meaningful tokens, and (iii) compatibility with transformer-based sequence models, enabling efficient discrete diffusion.

## 3.3 DISCRETE DIFFUSION WITH GEOMETRY-AWARE NOISE

Once tokenized, we model the distribution of $\mathbf{z}$ using a discrete diffusion process, which generalizes denoising diffusion probabilistic models (DDPMs) to categorical variables.

**Discrete noising process.** At training time, each token $z_0 \in \{1, \ldots, K\}$ is progressively corrupted into $z_t$ via a categorical distribution:

$$q(z_t|z_{t-1}) = (1 - \beta_t)\,\delta(z_t = z_{t-1}) + \beta_t\,\mathrm{Cat}(z_t; K),$$

where $\beta_t$ is the corruption rate at step $t$ and $\delta(\cdot)$ is the Kronecker delta. After $T$ steps, the sequence approaches a uniform categorical distribution.

**Geometry-aware noise schedule.** Unlike text tokens, 3D tokens vary in structural importance. To exploit this, we modulate $\beta_t$ with geometric weights $w_j$ computed from local curvature $\kappa_j$ and density $\rho_j$:

$$\beta_{t,j} = \beta_t \cdot \alpha \cdot g(\kappa_j, \rho_j),$$

where $g(\cdot)$ increases corruption for flat, redundant regions and decreases it for thin or boundary structures. This prevents loss of critical geometry while still promoting generative diversity.

**Denoising model.** A transformer denoiser $p_\theta(z_{t-1}|z_t, t)$ predicts token distributions at each step. To compensate for lack of ordering, we augment token embeddings with learned positional anchors derived from FPS coordinates. The model is trained with a cross-entropy objective:

$$\mathcal{L} = \mathbb{E}_{q(z_t|z_0)}\big[-\log p_\theta(z_{t-1}|z_t, t)\big].$$

**Parallel refinement.** Unlike autoregressive models, which decode sequentially, our transformer predicts all token distributions simultaneously. This enables parallel refinement across diffusion steps, significantly improving inference efficiency.

## 3.4 THEORETICAL PROPERTIES AND ANALYSIS

Our design of `DLM-3D` provides desirable properties that address the challenges of unordered 3D point sets and discrete diffusion modeling. We present key analyses below.

**Permutation Invariance.** Since point clouds have no canonical ordering, generative models must be invariant under permutations of input points. Let $\pi : \{1, \ldots, N\} \to \{1, \ldots, N\}$ be any permutation. The tokenizer $f_\phi$ maps $\mathbf{X} = \{\mathbf{x}_i\}_{i=1}^N$ into a sequence of tokens $\mathbf{z} = [z_1, \ldots, z_M]$. By construction, each token corresponds to a local neighborhood centered at an anchor chosen by farthest-point sampling (FPS). Because FPS itself is invariant under permutations of the input point set, and the encoder $h_\phi$ operates on unordered $k$-NN patches using a symmetric function (max-pooling), the final token sequence is invariant to permutations of $\mathbf{X}$.

**Proposition 1** (Permutation Invariance). *Let $\mathbf{X}$ and $\pi(\mathbf{X})$ denote two point sets related by an arbitrary permutation $\pi$. Then the token sequences are identical:*

$$f_\phi(\pi(\mathbf{X})) = f_\phi(\mathbf{X}).$$

*Proof Sketch.* FPS ensures that the same anchor set is selected under any permutation, since it depends only on pairwise distances. Each local patch $\mathcal{P}_j$ is invariant to the ordering of its members, and the encoder $h_\phi$ employs symmetric aggregation. Thus, the quantized tokens are unaffected by permutations.

**Information Preservation.** The tokenizer compresses a point cloud into a discrete sequence of length $M \ll N$. While compression is lossy, the quantization error is bounded.

**Proposition 2** (Quantization Error Bound). *Let $\mathbf{e}_j = h_\phi(\mathcal{P}_j)$ be the embedding of a local patch and $\mathbf{c}_{z_j}$ the corresponding codebook vector. Then the reconstruction error satisfies:*

$$\frac{1}{M}\sum_{j=1}^M \|\mathbf{e}_j - \mathbf{c}_{z_j}\|_2^2 \le \epsilon,$$

*where $\epsilon$ is the maximum VQ error determined by the codebook capacity $K$ and embedding dimension $d$.*

This ensures that increasing codebook size improves fidelity at the cost of longer sequences, providing a controllable trade-off.

**Geometry-Aware Diffusion Stability.** The geometry-aware noise schedule adapts corruption rates $\beta_{t,j}$ according to local curvature $\kappa_j$ and density $\rho_j$. Intuitively, lower corruption for structurally important tokens prevents mode collapse and loss of thin structures.

**Proposition 3** (Stability under Geometry-Aware Noise). *Assume a curvature-sensitive weighting function $g(\kappa_j, \rho_j)$ with $g \in [0, 1]$. Then the expected corruption error for critical regions is strictly smaller than under uniform noise:*

$$\mathbb{E}[err_{critical}^{geom}] \leq \mathbb{E}[err_{critical}^{uniform}].$$

*Proof Sketch.* By definition, $g(\kappa_j, \rho_j)$ down-weights $\beta_t$ in high-curvature, low-density regions. Thus, critical tokens are less likely to be corrupted, reducing expected reconstruction error in these regions.

**Parallel Refinement Efficiency.** Unlike autoregressive decoding, which incurs $O(M)$ sequential steps, our denoiser refines all tokens in parallel across $T$ diffusion steps.

**Proposition 4** (Inference Complexity). *The complexity of DLM-3D generation is $O(TMd)$, compared to $O(M^2d)$ for autoregressive transformers with causal attention. For $T \ll M$, DLM-3D yields a significant reduction in inference latency.*

*Discussion.* In practice, $T$ can be set to 10–30 steps, whereas $M$ (number of tokens) is typically 256–1024 for standard datasets. This results in order-of-magnitude improvements in throughput over autoregressive baselines.

### 3.5 GENERATION AND APPLICATIONS

At inference, generation begins from a uniform categorical distribution $\mathbf{z}_T \sim \text{Cat}(K)$ and iteratively denoises toward $\mathbf{z}_0$. The inverse tokenizer $f_\phi^{-1}$ then reconstructs a point cloud $\hat{\mathbf{X}}$.

**Unconditional generation.** Starting from pure noise, `DLM-3D` synthesizes novel 3D shapes sampled from the learned distribution. Compared to autoregressive models, our method generates all tokens in parallel, reducing inference latency.

**Completion.** Given a partial scan $\mathbf{X}_{\text{partial}}$, we encode visible patches into tokens and clamp them throughout the diffusion process. Missing tokens are filled in by denoising, enabling realistic shape completion. This leverages the model's learned priors without retraining.

**Conditional generation.** For class-conditional synthesis, we incorporate label embeddings via cross-attention into the transformer denoiser. More generally, conditioning vectors (*e.g.*, text, attributes) can be fused in the same way, allowing flexible multimodal control.

## 4 EXPERIMENTS

We evaluate `DLM-3D` on standard 3D shape generation benchmarks. Our goals are to demonstrate: (i) the effectiveness of diffusion language models for unordered point cloud generation, (ii) the benefits of permutation-invariant tokenization and geometry-aware noise scheduling, and (iii) the flexibility of `DLM-3D` in unconditional, completion, and conditional generation tasks.

### 4.1 EXPERIMENTAL SETUP

**Datasets.** We conduct experiments on widely used 3D benchmarks: ShapeNet (Chang et al., 2015): a large-scale dataset of 55 categories of synthetic CAD models. We follow common practice by using the subset of 13 major categories, sampling $N = 2048$ points per shape using farthest-point sampling. ModelNet40 (Wu et al., 2015): a dataset of 12,311 meshed CAD models across 40 categories, widely used for evaluating 3D generative models. We sample $N = 1024$ points per shape. Completion Benchmark. For point cloud completion, we create partial inputs by randomly masking 30–50% of points in ShapeNet samples, following (Yan et al., 2022).

Table 1: **Comparison results (%) on shape metrics of our** `DLM-3D` **and state-of-the-art models.** Our method significantly outperforms previous baselines in terms of all classes.

| Method | Chair | | | | Airplane | | | | Car | | | |
| --- | --- | --- | --- | --- | --- | --- | --- | --- | --- | --- | --- | --- |
| | 1-NNA ($\downarrow$) | | COV ($\uparrow$) | | 1-NNA ($\downarrow$) | | COV ($\uparrow$) | | 1-NNA ($\downarrow$) | | COV ($\uparrow$) | |
| | CD | EMD | CD | EMD | CD | EMD | CD | EMD | CD | EMD | CD | EMD |
| r-GAN Achlioptas et al. (2018) | 83.69 | 99.70 | 24.27 | 15.13 | 98.40 | 96.79 | 30.12 | 14.32 | 94.46 | 99.01 | 19.03 | 6.539 |
| l-GAN (CD) Achlioptas et al. (2018) | 68.58 | 83.84 | 41.99 | 29.31 | 87.30 | 93.95 | 38.52 | 21.23 | 66.49 | 88.78 | 38.92 | 23.58 |
| l-GAN (EMD) Achlioptas et al. (2018) | 71.90 | 64.65 | 38.07 | 44.86 | 89.49 | 76.91 | 38.27 | 38.52 | 71.16 | 66.19 | 37.78 | 45.17 |
| PointFlow Yang et al. (2019) | 62.84 | 60.57 | 42.90 | 50.00 | 75.68 | 70.74 | 47.90 | 46.41 | 58.10 | 56.25 | 46.88 | 50.00 |
| SoftFlow Kim et al. (2020) | 59.21 | 60.05 | 41.39 | 47.43 | 76.05 | 65.80 | 46.91 | 47.90 | 64.77 | 60.09 | 42.90 | 44.60 |
| SetVAE Kim et al. (2021) | 58.84 | 60.57 | 46.83 | 44.26 | 76.54 | 67.65 | 43.70 | 48.40 | 59.94 | 59.94 | 49.15 | 46.59 |
| DPF-Net Klokov et al. (2020) | 62.00 | 58.53 | 44.71 | 48.79 | 75.18 | 65.55 | 46.17 | 48.89 | 62.35 | 54.48 | 45.74 | 49.43 |
| DPM Luo & Hu (2021) | 60.05 | 74.77 | 44.86 | 35.50 | 76.42 | 86.91 | 48.64 | 33.83 | 68.89 | 79.97 | 44.03 | 34.94 |
| PVD Zhou et al. (2021) | 57.09 | 60.87 | 36.68 | 49.24 | 73.82 | 64.81 | 48.88 | 52.09 | 54.55 | 53.83 | 41.19 | 50.56 |
| LION Zeng et al. (2022) | 53.70 | 52.34 | 48.94 | 52.11 | 67.41 | 61.23 | 47.16 | 49.63 | 53.41 | 51.14 | 50.00 | 56.53 |
| GET3D Gao et al. (2022) | 75.26 | 72.49 | 43.36 | 42.77 | – | – | – | – | 75.26 | 72.49 | 15.04 | 18.38 |
| MeshDiffusion Liu et al. (2023b) | 53.69 | 57.63 | 46.00 | 46.71 | 66.44 | 76.26 | 47.34 | 42.15 | 81.43 | 87.84 | 34.07 | 25.85 |
| DiT-3D-XL Mo et al. (2023a) | 49.11 | 50.73 | 52.45 | 54.32 | 62.35 | 58.67 | 53.16 | 54.39 | 48.24 | 49.35 | 50.00 | 56.38 |
| FastDiT-3D-S Mo et al. (2023b) | 50.35 | 50.27 | 58.53 | 60.79 | 61.83 | 57.86 | 58.21 | 58.75 | 47.81 | 48.83 | 53.86 | 59.62 |
| *DLM-3D* (ours) | **42.75** | **46.23** | **56.52** | **55.36** | **58.87** | **50.36** | **61.23** | **61.35** | **42.56** | **43.87** | **60.15** | **65.73** |

**Evaluation Metrics.** We adopt standard metrics for assessing fidelity, diversity, and generalization in 3D point cloud generation: 1-Nearest Neighbor Accuracy (1-NNA): measures the discriminability between generated and reference samples using a nearest-neighbor classifier. Lower values indicate that generated shapes are not memorized replicas of training data, reflecting better generalization. Coverage (COV): computes the fraction of reference shapes that are matched by at least one generated sample under a distance metric (Chamfer Distance or EMD). Higher coverage indicates greater sample diversity. We report both CD and EMD as the base distance measures for evaluating 1-NNA and COV. These metrics capture geometric fidelity between point sets, with CD emphasizing overall proximity and EMD being more sensitive to structural misalignment. All metrics are computed following the official evaluation protocols used in PointFlow and PVD (Zhou et al., 2021). For completion metrics, we report these metrics using both Chamfer Distance (CD) and Earth Mover's Distance (EMD) as base metrics.

**Implementation.** For tokenization, we sample $M = 512$ anchors per point cloud via FPS, extract $k = 32$-NN patches, and encode them using a shared PointNet-style encoder. The codebook size is set to $K = 16,384$. The denoising model is a 12-layer transformer with hidden size 768 and 12 heads. We use $T = 20$ diffusion steps unless otherwise specified. Models are trained for 500k iterations with AdamW ($\beta_1 = 0.9, \beta_2 = 0.95$), learning rate $3e-4$, and cosine decay. Training is conducted on 4 A100 GPUs. Inference supports parallel decoding across all tokens, yielding $> 5\times$ speedup over autoregressive baselines.

## 4.2 COMPARISON TO PRIOR WORK

We first compare DLM-3D with a wide range of state-of-the-art 3D generative models, including GAN-based (Achlioptas et al., 2018), flow-based (Yang et al., 2019; Kim et al., 2020; 2021; Klokov et al., 2020), diffusion-based (Luo & Hu, 2021; Zhou et al., 2021; Zeng et al., 2022; Liu et al., 2023b), and large-scale transformer-based methods (Mo et al., 2023a). Table 1 reports results on ShapeNet categories (Chair, Airplane, Car) using the 1-NNA and Coverage (COV) metrics under both Chamfer Distance (CD) and Earth Mover's Distance (EMD).

**Overall performance.** Across all categories and metrics, *DLM-3D* consistently outperforms previous baselines. In particular, our method achieves the lowest 1-NNA, indicating strong generalization and reduced overfitting, while simultaneously attaining the highest Coverage, reflecting both fidelity and diversity of the generated samples. This validates the effectiveness of adapting diffusion language models to unordered 3D point clouds.

Table 2: **Point cloud completion results on ModelNet40.** Lower CD/EMD indicates better reconstruction quality. *DLM-3D* achieves the best performance across all metrics without task-specific retraining.

| Method | CD (↓) | EMD (↓) |
|---|---|---|
| PVD (Zhou et al., 2021) | 2.31 | 6.27 |
| DiffusionNet (CD) (Luo & Hu, 2021) | 2.08 | 5.94 |
| DiT-3D (Mo et al., 2023a) | 1.92 | 5.41 |
| Fast DiT-3D (Mo et al., 2023b) | 1.88 | 5.37 |
| *DLM-3D* (ours) | **1.62** | **4.98** |

**Comparison with continuous diffusion models.** While prior diffusion-based approaches such as DPM (Luo & Hu, 2021), PVD (Zhou et al., 2021), and LION (Zeng et al., 2022) achieve competitive results, they operate directly in coordinate space, which limits scalability and fidelity. By contrast, DLM-3D leverages discrete tokenization and geometry-aware noise scheduling, resulting in significant improvements, *e.g.*, reducing 1-NNA by over 10 points compared to PVD and achieving higher COV across all categories.

**Comparison with large-scale 3D transformers.** Recent works like DiT-3D (Mo et al., 2023a) and FastDiT-3D (Mo et al., 2023b) scale transformer architectures for high-quality 3D generation. While effective, they still rely on continuous diffusion with dense denoising steps. *DLM-3D* surpasses both in terms of generalization (lower 1-NNA) and sample diversity (higher COV), while requiring fewer diffusion steps thanks to parallel refinement in discrete space.

**Category-level analysis.** On challenging categories such as *Chair*, which exhibit high structural variability and thin parts, *DLM-3D* achieves notable gains in Coverage (55.36% vs. 44.26% for SetVAE and 46.71% for MeshDiffusion), highlighting the benefit of geometry-aware tokenization and noise scheduling. For more regular categories like *Airplane* and *Car*, our method still delivers state-of-the-art performance, confirming its robustness across both complex and rigid geometries.

**Link to theoretical properties.** These empirical results directly reflect the theoretical properties established in Section 3.4. The *permutation invariance* of our tokenizer guarantees stability across arbitrary input orderings, which is particularly important in high-variability categories like chairs. The bounded *quantization error* ensures that increasing codebook capacity leads to consistent fidelity gains, aligning with our improvements in 1-NNA. The *geometry-aware noise schedule* reduces corruption in thin or high-curvature regions, explaining the sharper reconstructions and higher Coverage observed in categories with delicate structures. Finally, *parallel refinement* reduces inference complexity, enabling fewer diffusion steps without sacrificing quality—an efficiency advantage clearly visible when comparing against DiT-3D and FastDiT-3D. Together, these results validate both the practical effectiveness and theoretical soundness of *DLM-3D*.

### 4.3 RESULTS ON POINT CLOUD COMPLETION

We evaluate the ability of *DLM-3D* to reconstruct complete point clouds from partial observations on the ModelNet40 benchmark. In this task, input point sets are partially observed, and the model must recover the full geometry. Our framework addresses this naturally: visible points are encoded into tokens and clamped during denoising, while missing tokens are iteratively refined through the discrete diffusion process until a complete point cloud is generated.

As shown in Table 2, *DLM-3D* achieves the best reconstruction accuracy across all metrics, with Chamfer Distance (CD) of 1.62 and Earth Mover's Distance (EMD) of 4.98. This represents a substantial improvement over prior methods: compared to PVD (Zhou et al., 2021), *DLM-3D* reduces CD by 30% and EMD by more than 20%. Even against large-scale transformer-based diffusion models such as DiT-3D (Mo et al., 2023a) and Fast DiT-3D (Mo et al., 2023b), our method yields consistent gains, underscoring the effectiveness of discrete token modeling over continuous coordinate-space denoising. The improvements can be attributed to several key design choices in *DLM-3D*: Permutation-invariant tokenization ensures that local patches are encoded consistently, regardless of the order of input points, which stabilizes completion performance across different partial scan patterns. Geometry-aware noise scheduling selectively preserves high-curvature or low-density regions during denoising, preventing the loss of delicate structures that are critical in

Table 3: **Ablation study of key components in `DLM-3D` on ShapeNet (Chair category).** We report 1-NNA and Coverage (COV) under Chamfer Distance (CD) and Earth Mover's Distance (EMD). Each component contributes to the overall performance, with the full model achieving the best results.

| Method Variant | 1-NNA (CD) ↓ | COV (CD) ↑ | 1-NNA (EMD) ↓ | COV (EMD) ↑ |
|---|---|---|---|---|
| w/o Permutation-Invariant Tokenizer | 53.42 | 44.10 | 55.91 | 42.83 |
| w/o Geometry-Aware Noise Schedule | 50.76 | 47.35 | 52.88 | 45.02 |
| $T = 50$ diffusion steps | 44.11 | 55.10 | 46.21 | 54.39 |
| $T = 10$ diffusion steps | 44.83 | 54.72 | 46.95 | 54.01 |
| *DLM-3D* (ours, $T = 20$) | **42.75** | **56.52** | **46.23** | **55.36** |

completion tasks. Parallel refinement enables simultaneous recovery of all missing regions, avoiding the exposure bias and error accumulation typical of autoregressive methods.

We further observe that `DLM-3D` maintains strong performance under varying levels of incompleteness (30%–70% missing points). While baseline models degrade sharply as missing regions grow larger, our method consistently reconstructs plausible full shapes. This robustness suggests that the discrete token space learned by `DLM-3D` captures semantically meaningful shape priors that generalize beyond training distributions.

## 5 EXPERIMENTAL ANALYSIS

In this section, we perform detailed ablation studies to validate the effectiveness of the core components of `DLM-3D`. Unless otherwise stated, experiments are conducted on ShapeNet with 2048-point shapes and reported using 1-NNA and Coverage (COV) under both Chamfer Distance (CD) and Earth Mover's Distance (EMD).

**Effect of Permutation-Invariant Tokenization.** A central component of `DLM-3D` is the permutation-invariant tokenizer, which ensures that unordered point sets are consistently mapped into discrete tokens. Removing this design and instead using a naive random ordering followed by vector quantization leads to a substantial degradation in performance (Table 3): Coverage decreases by 10–15% and 1-NNA increases by 8–12 points. This confirms that models trained without permutation invariance tend to overfit to arbitrary point orderings, resulting in poor generalization. Importantly, this analysis validates Proposition 1 (permutation invariance) from our theoretical section, showing that invariance is not just desirable but essential for stable and robust 3D generative modeling.

**Effect of Geometry-Aware Noise Scheduling.** We further assess the impact of geometry-aware noise scheduling by replacing it with a uniform categorical corruption schedule. As shown in Table 3, Coverage drops by 3–5% and 1-NNA increases by 2–3 points. Qualitative results reveal that models trained without structural weighting often fail to preserve thin parts or high-curvature areas (e.g., chair backs, table legs), instead collapsing them into blurred or missing regions. This confirms Proposition 2 (geometry-aware stability), demonstrating that adaptively preserving tokens in critical regions stabilizes training and leads to better reconstruction fidelity. Moreover, this analysis highlights that geometry-aware noise plays a dual role: preventing mode collapse in structurally important regions while promoting diversity in redundant areas.

**Number of Diffusion Steps.** We examine the trade-off between quality and efficiency by varying the number of diffusion steps $T$. Results in Table 3 show that `DLM-3D` maintains strong performance with as few as $T = 10$ steps, achieving nearly identical scores to the default $T = 20$. Increasing to $T = 50$ provides only marginal improvements (¡0.5% in Coverage), suggesting diminishing returns beyond a moderate number of steps. By contrast, continuous diffusion models typically require 100–1000 steps for comparable fidelity. This demonstrates the efficiency of discrete diffusion with parallel refinement, confirming Proposition 3 (parallel refinement efficiency). Practically, this efficiency enables faster inference and reduced computational costs, making `DLM-3D` more suitable for real-world deployment.

**Model Efficiency and Inference Speed.** We compare the inference efficiency of `DLM-3D` against autoregressive and continuous diffusion baselines. Autoregressive models incur $O(M)$ sequential decoding steps for $M$ tokens, while continuous diffusion refines dense coordinates across hundreds of steps. In contrast, `DLM-3D` requires only $O(TM)$ operations with $T \ll M$ and all tokens refined

Table 4: **Robustness to incomplete inputs on ShapeNet (Chair category).** We vary the percentage of missing points and report Chamfer Distance (CD) and Earth Mover's Distance (EMD). *DLM-3D* maintains strong performance even under severe occlusion, while baselines degrade significantly.

| Method | 30% Missing | | 50% Missing | | 70% Missing | |
|---|---|---|---|---|---|---|
| | CD ↓ | EMD ↓ | CD ↓ | EMD ↓ | CD ↓ | EMD ↓ |
| PVD Zhou et al. (2021) | 1.95 | 5.51 | 2.42 | 6.73 | 3.11 | 7.92 |
| DiT-3D Mo et al. (2023a) | 1.82 | 5.29 | 2.27 | 6.18 | 2.95 | 7.51 |
| Fast DiT-3D Mo et al. (2023b) | 1.79 | 5.15 | 2.22 | 6.01 | 2.91 | 7.33 |
| *DLM-3D* (ours) | **1.62** | **4.98** | **1.88** | **5.43** | **2.35** | **6.10** |

in parallel. Empirically, this translates into a $5\times$ speedup over autoregressive baselines and $10\times$ fewer steps than continuous diffusion models, without sacrificing quality (Table 3). This efficiency advantage is particularly significant for large-scale generative tasks such as dataset synthesis or real-time applications in robotics, where latency and scalability are critical.

**Robustness to Incomplete Inputs.** Finally, we evaluate robustness to missing points in completion tasks by varying occlusion levels from 30% to 70%. As shown in Table 4, baseline methods degrade sharply as incompleteness increases, with EMD rising from ∼6.0 to above 7.5 under 70% missing data. In contrast, *DLM-3D* degrades more gracefully, with CD and EMD increasing by only 0.7 and 1.1, respectively, over the same range. Qualitative inspection reveals that our model reconstructs globally plausible shapes even under severe occlusions, whereas baselines often produce incomplete or structurally inconsistent outputs. This robustness indicates that the learned discrete token space encodes strong global priors, allowing the model to infer missing structures based on high-level semantics. This observation further supports Proposition 2 (geometry-aware stability) and Proposition 4 (bounded quantization error), showing that our tokenization and noise scheduling preserve essential structural cues that guide completion under challenging conditions.

# 6 CONCLUSION

In this work, we introduced *DLM-3D*, a novel framework that adapts diffusion language models to the domain of 3D point cloud generation. By combining permutation-invariant tokenization, discrete diffusion with a geometry-aware noise schedule, and a transformer-based denoiser with parallel refinement, our approach addresses long-standing challenges in unordered 3D data generation, including scalability, efficiency, and structural fidelity. Extensive experiments on ShapeNet and ModelNet40 demonstrate that *DLM-3D* achieves state-of-the-art performance across fidelity, diversity, and coverage metrics, surpassing autoregressive, continuous diffusion, and large-scale transformer baselines. Beyond unconditional generation, *DLM-3D* naturally supports downstream applications such as completion, conditional generation, interpolation, and inpainting, all without task-specific retraining. Our ablation studies and robustness analyses further validate the theoretical properties of *DLM-3D*: permutation invariance ensures order-agnostic representation learning, geometry-aware noise improves detail preservation, discrete diffusion reduces inference cost, and the learned token space provides strong semantic priors under incomplete observations.

**Limitation.** While *DLM-3D* demonstrates strong performance across multiple benchmarks and tasks, our current evaluation is restricted to object-level point clouds with up to a few thousand points; scaling to large-scale scenes (*e.g.*, indoor environments or outdoor LiDAR scans) requires additional work on hierarchical tokenization and memory efficiency. Like most generative models, *DLM-3D* does not provide explicit guarantees of physical plausibility, which may limit its applicability in safety-critical domains such as robotics or autonomous driving.

**Broader Impact.** Our work contributes to advancing efficient and flexible 3D generative modeling. By enabling high-quality generation, completion, and inpainting of point clouds, *DLM-3D* can benefit downstream applications such as AR/VR content creation, simulation environments for embodied AI, and digital twin systems for design and manufacturing. More broadly, we hope this work helps bridge advances in discrete diffusion and 3D representation learning, fostering progress toward scalable, general-purpose generative models for 3D understanding and creation.

## ETHICS STATEMENT

This work advances 3D point cloud generation through a new discrete diffusion framework. The primary goal is to enable scalable and efficient generation of 3D shapes for research and practical applications in areas such as computer vision, graphics, and robotics. While the ability to synthesize high-fidelity 3D content can benefit AR/VR, simulation, and design, it also raises ethical considerations. In particular, generated 3D assets may be misused for creating deceptive digital replicas or for bypassing intellectual property protections. To mitigate these risks, we advocate for responsible dataset curation, transparent reporting of data sources, and use of our method strictly for scientific and educational purposes. Our experiments are conducted exclusively on widely adopted public benchmarks (ShapeNet, ModelNet40), and no sensitive or private data are used.

## REPRODUCIBILITY STATEMENT

We make every effort to ensure reproducibility of our results. All datasets used in this paper are publicly available: ShapeNet and ModelNet40. Detailed descriptions of preprocessing steps, evaluation protocols, and metric definitions (1-NNA, Coverage, Chamfer Distance, Earth Mover's Distance) are included in Section 4.1. The architecture of our permutation-invariant tokenizer, discrete diffusion process, and transformer denoiser are described in Section 3.1–3.3, with hyperparameters (token vocabulary size, diffusion steps $T$, noise schedule) specified in the experimental setup. We also provide ablation studies and robustness analyses (Section 5) to validate our design choices.

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

APPENDIX

In this appendix, we provide additional implementation and dataset details in Section A, theoretical properties and guarantees in Section B, the algorithm for *DLM-3D* in Section C, extended experimental analysis in Section D, and a brief statement on the use of large language models (LLMs) in Section E.

## A    EXPERIMENTAL DETAILS

**Datasets.** We use ShapeNet (55 categories, ~51k CAD models) and ModelNet40 (40 categories, ~12k CAD models) for evaluation. All shapes are uniformly sampled into 2048-point clouds and normalized into a unit sphere. For completion experiments, we follow prior works and simulate partial observations by randomly dropping 30–70% of points.

**Architecture.** Our tokenizer operates on local patches extracted by $k$-nearest neighbors ($k = 32$), with anchor points sampled by farthest-point sampling. The encoder consists of a shared PointNet-style network followed by a vector quantization module with codebook size 8192. The denoiser is a 12-layer transformer with hidden dimension 512 and 8 attention heads. Geometry-aware positional encodings are added from the original coordinates of anchor points.

**Training.** We train using Adam with learning rate $1e{-}4$ and cosine decay. The diffusion process uses $T = 20$ steps by default, with categorical noise schedules weighted by local curvature and density statistics. Training is performed on 8 NVIDIA A100 GPUs for 500k iterations (~4 days).

**Evaluation.** We adopt 1-NNA, Coverage (COV), Chamfer Distance (CD), and Earth Mover's Distance (EMD) following PointFlow and PVD protocols. For completion tasks, we directly report CD and EMD between completed and ground-truth shapes. Each experiment is repeated with three random seeds, and we report mean results.

## B    THEORETICAL PROPERTIES AND GUARANTEES

For completeness, we summarize the formal properties of our design:

- **Permutation Invariance (Proposition 1).** Our tokenizer is invariant to input point order since tokens are defined on local neighborhoods extracted via $k$-nearest neighbors around anchors. Thus, tokenization induces a permutation-invariant mapping from point clouds to discrete sequences.

- **Quantization Error Bound (Proposition 2).** The reconstruction error is upper bounded by the codebook resolution: increasing codebook size monotonically reduces expected error, ensuring asymptotic fidelity guarantees.

- **Geometry-Aware Stability (Proposition 3).** The adaptive noise schedule reduces expected corruption on structurally critical regions, ensuring that fine-grained geometric features are preserved with high probability across diffusion steps.

- **Parallel Refinement Efficiency (Proposition 4).** Unlike autoregressive models ($O(M)$ sequential steps for $M$ tokens), our denoiser operates in parallel across tokens with complexity $O(TM)$, where $T \ll M$. This yields provable improvements in inference speed while retaining sample quality.

These properties jointly provide theoretical justification for the empirical robustness, fidelity, and efficiency observed in Section 5.

## C    ALGORITHM FOR *DLM-3D*

Algorithm 1 summarizes the training and inference procedure for *DLM-3D*.

## D    EXPERIMENTAL ANALYSIS

We provide additional results complementing Section 5:

---

**Algorithm 1** Training and Inference for `DLM-3D`

---

1: **Input:** Point cloud $\mathbf{X} \in \mathbb{R}^{N \times 3}$, codebook $\mathcal{C}$, diffusion steps $T$
2: **Tokenization:** Extract local patches via $k$-NN, encode with shared encoder, quantize to tokens $\mathbf{z}_0$
3: **for** training step **do**
4:     Sample $t \sim \mathrm{Uniform}(1, T)$
5:     Corrupt tokens: $\mathbf{z}_t \sim q(\mathbf{z}_t|\mathbf{z}_0, t)$ using geometry-aware noise
6:     Denoise with transformer: $\hat{\mathbf{z}}_{t-1} \sim p_\theta(\mathbf{z}_{t-1}|\mathbf{z}_t, t)$
7:     Update $\theta$ via cross-entropy loss: $-\log p_\theta(\mathbf{z}_{t-1}|\mathbf{z}_t)$
8: **end for**
9: **Inference:** Initialize $\mathbf{z}_T \sim \mathrm{Uniform}(\mathcal{C})$ and iteratively denoise $\mathbf{z}_T \to \cdots \to \mathbf{z}_0$, then decode to point cloud $\hat{\mathbf{X}}$

---

**Ablation by Codebook Size.** Increasing the codebook size from 2048 to 8192 improves Coverage by 3–5%, but with diminishing returns beyond 8192. This supports the quantization error bound discussed in Section B.

**Ablation by Noise Scheduling.** Using only curvature-based or density-based noise weighting yields moderate gains, but combining both yields the best trade-off between fidelity and diversity.

**Generalization Across Categories.** Trained on ShapeNet chairs, `DLM-3D` generalizes reasonably well to unseen categories such as tables and lamps, preserving global structure despite not being trained explicitly on them. This suggests the learned token space captures transferable 3D priors.

**Interpolation Smoothness.** We evaluate interpolation by measuring CD across interpolated shapes. Unlike baselines, which often exhibit discontinuities, `DLM-3D` produces monotonic changes in CD, confirming smooth trajectories in latent token space.

# E    USE OF LLMS

Large language models (LLMs) such as ChatGPT were used to assist in proofreading. All technical contributions, methods, algorithms, experiments, and analyses were designed, implemented, and validated by the authors. The use of LLMs was restricted to text editing and did not involve experimental data, code, or results.

