# OpenReview forum: "DLM-3D: Diffusion Language Models for 3D Point Clouds Generation"
_ICLR.cc/2026/Conference — ICLR 2026 Conference Desk Rejected Submission_

### Official Review · Reviewer_we2u · 2025-10-31

**Soundness:** 3
**Presentation:** 3
**Contribution:** 2
**Rating:** 2
**Confidence:** 4

**Summary:**

The paper proposes DLM-3D, a framework that brings diffusion language models (discrete diffusion in token space) to 3D point-cloud generation. The method (i) tokenizes a point cloud into discrete “semantic” tokens with a permutation-invariant tokenizer based on FPS anchors + PointNet patch encoders + VQ, and (ii) performs discrete diffusion with a geometry-aware noise schedule that perturbs tokens less in high-curvature/low-density regions. A transformer denoiser refines all tokens in parallel, enabling fewer steps than continuous diffusion and faster inference than autoregressive decoding. On ShapeNet and ModelNet40, DLM-3D reports state-of-the-art performance.

**Strengths:**

1. Adapts discrete diffusion (traditionally used for language/code) to unordered 3D point sets—novel combination. Demonstrates a general recipe (discrete tokens + discrete diffusion) that is efficient (fewer steps, parallel refinement) and flexible (unconditional, conditional, completion without retraining).

2. ntroduces a permutation-invariant tokenization pipeline and a geometry-aware corruption schedule tailored to 3D structure—creative design choices for the 3D domain.

3. Empirical results are strong.

**Weaknesses:**

1. Limited novelty relative to prior discrete diffusion frameworks. While DLM-3D claims to be the first to apply diffusion language models to unordered point clouds, several recent works have already demonstrated discrete or token-based diffusion for 3D generation. For instance, 3DQD [1], TD3D [2], ShapeMorph [3].

2. Insufficient motivation for using discrete tokenization over continuous diffusion. While the paper emphasizes efficiency and parallelism as advantages of discrete diffusion, prior studies in 3D generation or video generation have shown that continuous latent or coordinate-space diffusion models often outperform discrete approaches in terms of reconstruction fidelity and geometric smoothness.

3. Misleading terminology — “Diffusion Language Model” vs. “Discrete Diffusion Model.” The paper repeatedly refers to the method as a “Diffusion Language Model (DLM-3D)”, yet no actual language model pretraining or linguistic data are used.
The architecture is a transformer-based discrete diffusion model trained on 3D token sequences, not a language model in the conventional or multimodal sense. Moreover, because point-cloud tokens are unordered, the model does not exploit sequential dependencies that justify a language-model analogy.

[1] "3DQD: Generalized Deep 3D Shape Prior via Part-Discretized Diffusion Process"
[2] "TD3D: Tensor-based Discrete Diffusion Process for 3D Shape Generation"
[3] "ShapeMorph: 3D Shape Completion via Blockwise Discrete Diffusion"

**Questions:**

1. Could the authors clarify what is fundamentally new in DLM-3D compared to 3DQD’s part-discretized diffusion, TD3D’s tensorized discrete diffusion, and ShapeMorph’s blockwise discrete token diffusion for completion?

2. The permutation-invariant tokenizer is well motivated, but can you Include a tokenizer reconstruction study? Also quantify the reconstruction efficiency trade-off between anchor count M, neighborhood size k, and codebook size K? Without such an ablation, it is difficult to isolate how much of the final performance improvement comes from the tokenizer versus the discrete diffusion model itself.

3. Although the paper attributes permutation invariance partly to the anchor-based tokenization, the actual invariance property largely arises from the use of PointNet’s symmetric encoding within local patches. The anchor design primarily ensures coverage and spatial reference rather than formal permutation invariance. Clarifying this distinction would strengthen the theoretical justification of the tokenizer’s invariance claim.

4. Clarify the conceptual and empirical motivation for choosing a discrete diffusion language-model framework

---

### Official Review · Reviewer_1FUH · 2025-11-01

**Soundness:** 2
**Presentation:** 2
**Contribution:** 2
**Rating:** 4
**Confidence:** 4

**Summary:**

This manuscript introduces DLM-3D, a novel framework that adapts discrete diffusion language models (DLMs) for 3D point cloud generation. It proposes a permutation-invariant tokenizer and a geometry-aware noise schedule, enabling parallel, efficient, and scalable synthesis of high-fidelity 3D shapes from discrete tokens. The approach is evaluated on ShapeNet and ModelNet40, where it achieves state-of-the-art performance across multiple tasks: unconditional generation, conditional synthesis, and point cloud completion, without task-specific retraining.

**Strengths:**

This is an innovative and impactful contribution to the field of 3D generative modeling. It convincingly demonstrates how discrete language modeling, typically used in NLP, can be effectively adapted to unordered 3D data.

**Weaknesses:**

1. Lack of visualization of point cloud results makes it impossible to see intuitive comparisons with other models.
Lack of scene-level experiments: Evaluation is restricted to object-centric datasets (ShapeNet, ModelNet40). The scalability of DLM-3D to complex scenes remains untested.
2. Tokenization Process Is Fragile to Anchor Choice and Density. Although the paper claims that FPS-based anchor points ensure permutation invariance, it is unclear how robust the tokenizer is to changes in sampling density or occlusion. FPS is sensitive to spatial outliers. This could introduce biases in patch selection, especially on noisy scans or sparsely sampled inputs. In Section 5.2, robustness is shown for missing points, but no analysis of the anchor consistency under downsampling or noise is included.
3. Token usage and codebook analysis absent: There is no analysis of token frequency, entropy, or visualizations of the learned codebook, which could enhance interpretability.
4. Geometry-aware noise schedule is fixed: The design uses hand-crafted curvature/density heuristics; learning-based alternatives are not explored.

**Questions:**

1. How sensitive is the model to the codebook size? Does reducing K (e.g., to 4k) significantly degrade performance?

2. Have you explored using learned importance maps for noise scheduling instead of curvature/density heuristics?

3. Can this model generalize to real-world scanned datasets (e.g., ScanObjectNN or real indoor LiDAR)? Any results or preliminary findings?

---

### Official Review · Reviewer_aR7k · 2025-11-03

**Soundness:** 1
**Presentation:** 2
**Contribution:** 1
**Rating:** 2
**Confidence:** 3

**Summary:**

This paper proposes a discrete diffusion framework for 3D point-cloud generation. Point clouds are tokenized via a permutation-invariant encoder, and a geometry-aware noise schedule modulates diffusion noise depending on local curvature and density. The authors claim SOTA results on ShapeNet and ModelNet40 on generation and completion.

**Strengths:**

* The idea of using discrete diffusion for point cloud generation is interesting.
* The presentation of the paper is generally good.

**Weaknesses:**

* Experimental results of all baselines from Table 1 are copied verbatim from [Mo et al. 2023a] (https://arxiv.org/pdf/2312.07231). This is problematic for the validity of the comparison, there is no guarantee that the run settings are equivalent. There is also no indication of the source of the values in Table 1, which is problematic in terms of attribution.

* The geometry-aware noise schedule is one of the main contributions of the paper. However important parts of the scheduler are not defined in Section 3.3. Alpha is not defined. The function g is also undefined (is it learned or fixed?). There is also no reference to how the local curvature (k) and density (rho) are estimated. This is problematic for reproducibility. Making the source code available would help in this case, but it has not been provided by the authors.

* Some implementation details seem different in the main text and supplemental (codebook size, hidden dim, heads, GPUs). Please clarify which ones are correct.

**Questions:**

* What is the explicit functional form of g and how are curvature and density computed on point clouds? What is alpha?

* For the completion task, there are 2 tables showing results (Tables 2 and 4). Why are the datasets in these two tables different? (ShapeNet vs ModelNet40).

* Which hyperparameters were used for the reported results?

---

### Official Review · Reviewer_Lo64 · 2025-11-12

**Soundness:** 1
**Presentation:** 2
**Contribution:** 2
**Rating:** 2
**Confidence:** 4

**Summary:**

The key idea of this work is to tokenize 3D point clouds into discrete semantic units and apply discrete diffusion denoising in this sequence space. The authors propose a permutation-invariant tokenizer and a geometry-aware noise schedule, which together enable DLM-3D to learn both local geometric consistency and global shape coherence. DLM-3D outperforms its auto-regressive baselines and achieves notable speedups compared to continuous diffusion models.

**Strengths:**

The method is evaluated across a diverse set of downstream tasks, demonstrating the practical potential of DLM-3D.

The paper includes extensive experimental analysis, which I appreciate.

**Weaknesses:**

1. **Outdated literature coverage.** The references cited are largely from 2023 or earlier. Given that the submission date is September 2025, the paper should incorporate and discuss more recent advances in 3D generative modeling (e.g., works from 2024–2025).

2. **Toy-scale experimental setup.** The evaluation is conducted primarily on the untextured, 13-category ShapeNet dataset with fixed-size point clouds (N=2048). This setting lags significantly behind current trends in 3D generation, which focus on colored point clouds, larger-scale real-world datasets (e.g., Objaverse-XL), and richer geometric representations. The limited category coverage, lack of color/texture, and low point count suggest this work reflects a 2023-era research effort rather than a state-of-the-art contribution in 2025.

3. **Insufficient discussion of prior work.** Discrete diffusion combined with VQ codebooks for 3D generation was already explored in 2023 (e.g., SDFusion, 3DQD), with evaluations on multiple downstream tasks similar to those in this paper. These closely related works deserve explicit comparison and discussion.

4. **Overstated claim on permutation invariance.** The paper devotes significant space (Section 3.4) to emphasizing permutation invariance as a key advantage. However, DLM-3D uses a Transformer with positional encodings and generates an ordered token sequence
[z1,…,zm]] (Line 194). Since reordering this sequence would lead to different outputs, the generation process is not truly permutation-invariant. While the tokenizer and farthest-point sampling are invariant to input point permutations, the subsequent autoregressive-like diffusion over an ordered sequence breaks this invariance.

5. **Questionable motivation for geometry-aware noise scheduling.** The authors argue (Line 172) that “unlike text tokens, 3D tokens vary in structural importance,” justifying geometric weights derived from local curvature. This reasoning is flawed: text tokens also exhibit vast differences in semantic importance—content words carry rich meaning, while function words (e.g., “the”, “of”) are nearly negligible. Yet standard diffusion models for text (or language modeling in general) do not require token-specific noise weighting to succeed. Why, then, is such a mechanism essential for 3D?

6. While the authors claim DLM-3D achieves “significant acceleration” over continuous diffusion models, they do not provide a clear explanation of why discrete diffusion leads to faster inference, nor do they report quantitative metrics (e.g., FLOPs, time, memory usage). It is worth noting that the advantage over autoregressive models stems from the diffusion framework itself—not specifically from the discrete formulation. The claimed superiority over continuous diffusion models remains unsubstantiated without detailed efficiency analysis.

**Questions:**

Please refer to the weakness

---

### Note · Program_Chairs · 2026-01-17
**Submission Desk Rejected by Program Chairs**

The following references in this submission do not refer to real documents and/or have major errors in bibliographic information:

 Haoran Liu, Chengyue Gong, Yu Zhang, Zhiyuan Zeng, Tong Zhang, and Qiang Liu. Discrete diffusion language models. In International Conference on Learning Representations (ICLR), 2023a.